# Limitations of the NTK for Understanding Generalization in Deep Learning

## Abstract

The "Neural Tangent Kernel" (NTK) (Jacot et al., 2018), and its empirical variants have been proposed as a proxy to capture certain behaviors of real neural networks. In this work, we study NTKs through the lens of scaling laws, and demonstrate that they fall short of explaining important aspects of neural network generalization. In particular, we demonstrate realistic settings where finite-width neural networks have significantly better data scaling exponents as compared to their corresponding empirical and infinite NTKs at initialization. This reveals a more fundamental difference between the real networks and NTKs, beyond just a few percentage points of test accuracy. Further, we show that even if the empirical NTK is allowed to be pre-trained on a constant number of samples, the kernel scaling does not catch up to the neural network scaling. Finally, we show that the empirical NTK continues to evolve throughout most of the training, in contrast with prior work which suggests that it stabilizes after a few epochs of training. Altogether, our work establishes concrete limitations of the NTK approach in understanding generalization of real networks on natural datasets.

## 1 Introduction

The seminal work of Jacot et al Jacot et al. (2018) introduced the "Neural Tangent Kernel" (NTK) as the limit of neural networks with widths approaching infinity. Since this limit holds provably under certain initializations, and kernels are more amenable to analysis than neural networks, the NTK promises to be a useful reduction to understand deep learning. Thus, it has initiated a rich research program to use the NTK to explain various behaviors of neural networks, such as convergence to global minima (Du et al., 2018; 2019), good generalization performance (Allen-Zhu et al., 2018; Arora et al., 2019a), implicit bias of networks (Tancik et al., 2020) as well as neural scaling laws (Bahri et al., 2021).

In addition to the infinite NTK, the *emperical NTK* — the kernel with features that are gradients of a finite-width neural network— can be a useful object to study, since it is an approximation to both the true neural network and the infinite NTK. This has also been studied extensively as a tool to understand deep learning (Fort et al., 2020; Long, 2021; Paccolat et al., 2021; Ortiz-Jiménez et al., 2021).

In this work, we probe the upper limits of this research program: we want to understand the extent to which understanding NTKs (empirical and infinite) can teach us about the success of neural networks. We study this question under the lens of scaling (Kaplan et al., 2020; Rosenfeld et al., 2019)—how performance improves as a function of samples and as a function of time— since the scaling is an important "signature" of the mechanisms underlying any learning algorithm. We thus compare the scaling of real networks to the scaling of NTKs in the following ways.

1. *Data scaling of initial kernel (Section 3)*: We show that both the infinite and empirical NTK (at initialization) can have worse data scaling exponents than neural networks, in realistic settings (Figure 1). We find that this is robust to various important hyperparameter changes such as learning rate (in the range used in practice), batchsize and optimization method.

2. *Width scaling of initial kernel (Section 3):* Since neural networks provably converge to the NTK at infinite width, we investigate why the scaling behavior differs at finite width. We show (Figure 2(b), 2(c)) realistic settings where as the width of the neural network increases

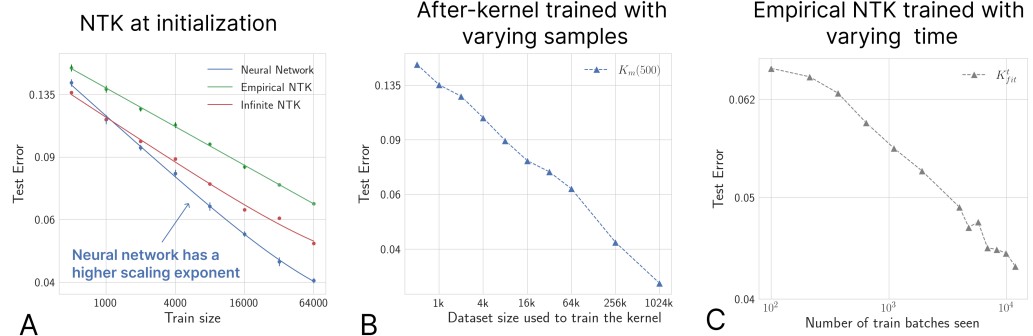

Figure 1: **Summary of results:** (A) *Neural network scales better than NTK at initialization:* We compare the scaling exponent of a neural network, its corresponding infinite and empirical NTK at initialization. Details in Section 3. (B) *After-kernel continues to improve with more training samples:* We train a neural network with $m = \{1K, 2K...1024K\}$ samples, extract the empirical NTK at completion, and use this kernel to fit 500 samples. Details in Section 4. (C) *Empirical NTK improves with constant rate with respect to training time:* We extract the empirical NTK at various times in training and use it to fit the full train dataset. Details in Section 5.

      to very large values, the test performance of the network gets *worse* and approaches the performance of the infinite NTK, unlike existing results in literature which suggest that increasing the width in the over-parameterized regime is always good. This also raises new questions about scaling of neural networks with width, in particular the "variance-limited" neural scaling regimes (Bahri et al., 2021).

3. *Data scaling of after-kernel (Section 4):* We consider the after-kernel (Long, 2021) i.e. empirical NTK extracted after training to completion on a fixed number of samples. We show (Figure 1(B), 4(b)) that the after-kernel continues to improve as we increase the training dataset size. On the other hand, we find (Figure 4(c)) that the scaling exponent of the after-kernel, extracted after training on a fixed number of samples remains *worse* than that of the corresponding neural network.

4. *Time scaling (Section 5):* We show (Figure 1(C), 5(a)) realistic settings where the empirical NTK continues to improve uniformly throughout most of the training. This is in contrast with prior work (Fort et al., 2020; Ortiz-Jiménez et al., 2021; Atanasov et al., 2021; Long, 2021) which suggests that the empirical NTK changes rapidly in the beginning of the training followed by a slowing of this change.

We demonstrate these phenomena occur in certain settings which are based on real, non-synthetic data, and modern architectures (for e.g.: for datasets CIFAR-10 and SVHN and convolutional networks). While we do not claim that these phenomena manifest for *all possible* datasets and architectures, we believe that our examples highlight important limitations to the use of NTK to understand the test performance of neural networks. Formalizing the set of distributions or architectures for which these phenomenon occur is an important direction for future theoretical research.

## 1.1 COMPARISON TO PRIOR WORK ON NTK GENERALIZATION

Our main focus is to understand feature learning occurring due to finite width. To do this, we make the following deliberate choices in all of our experiments: a) We use the NTK parameterization, this makes sure that infinite width networks will be equivalent to kernels b) We use the same optimization setup for the neural network, empirical NTK and Infinite NTK, this makes sure that as width tends to infinity all 3 models will have the same limit. We make sure that our comparisons are robust by c) using scaling laws to compare these models and d) doing various hyperparameter ablations (Figure 3).

Below we describe several lines of related works and how our work differs from them.

**Small initialization and representation learning at infinite width.** Infinite widths neural networks in the NTK and standard initialization are equivalent to kernels (Jacot et al., 2018; Yang & Hu, 2021). On the other hand it has been shown (Yang & Hu, 2021; Sirignano & Spiliopoulos, 2019; Nguyen & Pham, 2020; Araújo et al., 2019; Fang et al., 2020) that with small initialization feature learning is possible at infinite width. The feature learning displayed in our experiments is not due to small initialization as we initialize our networks in the NTK parameterization. This was a deliberate choice as we are interested in feature learning occurring due to finite width as this is the kind of feature learning displayed by empirical neural networks (which usually do not have a small initialization).

**Data Scaling for NTKs and neural networks.** Scaling laws have been empirically shown Kaplan et al. (2020); Rosenfeld et al. (2019) for neural networks and have been theoretically proven Bordelon et al. (2020); Canatar et al. (2021); Bordelon & Pehlevan (2022) for NTKs under natural assumptions. Comparison between the scaling laws for neural network and empirical NTKs has been previous looked at by Paccolat et al. (2021) and Ortiz-Jiménez et al. (2021) and both find that neural networks have better scaling than empirical NTK at initialization. Both of these papers do not compare to infinite NTKs which leaves open the possibility that neural networks and infinite width NTKs behave the same wrt their scaling constants.

**Pointwise comparisons of neural networks and corresponding infinite NTKs** has also been studied extensively in the literature (Arora et al., 2019b; Lee et al., 2020; Simon et al., 2021) but the results have been divided. As discussed earlier we focus on comparing scaling laws. We argue that scaling laws, instead of point-wise comparisons, are the appropriate tool to compare neural networks and NTKs. Practically, pointwise comparisons between any two models can be fraught with issues as the ordering can flip depending on dataset size, as well as the specific choice of hyperparameters. On the other hand, scaling exponents have been found to be more robust to the choice of hyperparameters (Bansal et al., 2022; Kaplan et al., 2020). More importantly, the claim that NTK can capture "most" of the performance of the neural network can be subjective, specially when we are comparing small error or loss values. We show that when we look closely at the scaling exponents of these objects instead, we find major differences.

**Theoretical studied effects of finite width with respect to the NTK regime.** Finite width corrections to the NTK theory have been studied by Andreassen & Dyer (2020); Roberts et al. (2021); Bahri et al. (2021). While these results do not need infinite widths they still require much higher than practically used widths particularly for the training sizes used in practice. These papers either consider a) the finite width corrections of empirical NTK or b) they consider the change in NTK but predict that the higher order analogues of empirical NTK remain constant. For a) we show that the empirical NTK is very far from the performance of finite width neural networks. Regarding b), in Appendix D we show that the higher order analogues of empirical NTK change significantly.

**After Kernel and Time Dynamics** We discuss these in detail in Section 4 and 5.

We describe other related works in Appendix G.

## 2 EXPERIMENTAL METHODOLOGY

Here we describe the common methodology used in our experiments.

The core object we want to understand is the data-scaling law of real neural networks— that is, what is its asymptotic performance as a function of the number of train samples? Concretely, in this work we restrict to classification problems, where we measure performance in terms of test classification error. For a given classification algorithm, let $L(n)$ be its *learning curve*: its expected test error as a function of number of samples $n$. In practice, many neural networks exhibit power-law decay in their learning curves (Kaplan et al., 2020). In such settings, we have $L(n) \sim \alpha n^\beta$ and we are interested primarily in the *scaling exponent* $\beta$, which determines the asymptotic rate of convergence.

**Empirical and Infinite NTK** Let $f(w, x)$ be a neural network with $w$ representing the weights and $x$ an input. By Taylor expansion around $w_0$ we have:

$$f(w,x) = f(w_0, x) + \nabla_w f(w,x)|_{w_0}(w - w_0) + \frac{1}{2}(w - w_0)^T \nabla_w^2 f(w,x)|_{w_0}(w - w_0) + \dots$$

Empirical NTK of the neural network around weights $w_0$ refers to the model $g_1(w, x) = \nabla_w f(w, x)|_{w_0}(w - w_0)$. Note that this is not the same as linearizing the network as we omit the $f(w_0, x)$ term. Empirical NTK is a linear model with respect to the weights $w$. Infinite NTK refers to the limit of the empirical NTK of the network around initial weights as width tends to infinity.

For a given learning problem and given neural network architecture NN, we want to understand its data-scaling law $L_{\text{NN}}(n)$. We consider the infinite NTK of the NN and the empirical NTK of the NN at initialization and their corresponding learning curves, $L_{\text{NTK}}(n)$ and $L_{\text{ENTK}}(n)$. Now we ask: is the scaling-exponent of $L_{\text{NN}}$ always close to the scaling-exponent of either $L_{\text{ENTK}}$ or $L_{\text{NTK}}$, in realistic settings? That is, how well does the NTK approximation capture the generalization of real networks, on natural distributions?

Recall that this question is especially interesting because the three objects involved (Neural Network, NTK, and ENTK) all become provably equivalent in the appropriate width $\to \infty$ limit. Thus, at infinite-width we know their scaling laws must be the equivalent. The question is then, how far are we from this limit in practice? Are the widths used in practice large enough for their scaling-behavior to be captured by the infinite-width limit? To probe these questions, we empirically study scaling laws of these methods on image-classification problems.

**Remark on comparisons.** We intentionally only compare a neural network to *its corresponding NTK*, and not to other kernels. Our motivation not address the question of "can (some) kernel perform as well as as a given neural network?"— indeed, there may be some better kernel to consider than the NTK. However, our goal is to study the specific kernels given by the NTK approximation, in correspondence with real networks.

**Datasets.** We use the following datasets:

1. A 2 class subset (dog, horse) of CIFAR-5m (Nakkiran et al., 2021) dataset, as a binary classification problem, which we denote *CIFAR-5m-bin*. This is a dataset of synthetic but realistic $32 \times 32$ RGB images similar to CIFAR-10, generated using a generative model.

2. A binary classification task on the SVHN dataset (Netzer et al., 2011) with the labels being the parity of the digit, denoted by *SVHN-parity*. For the training data we use a balanced random subset of 'train' and 'extra' partitions, for test data we use the 'test' partition.

We focus on the CIFAR-5m-bin experiments in the main body. Corresponding SVHN-parity experiments can be found in Appendix F.

We use these particular datasets because we need datasets with a large number of samples in order to measure data-scaling, and CIFAR-5m-bin and the SVHN dataset both have $\geq 600$k samples. We chose to consider binary tasks as this makes the kernel experiments computationally feasible. Although there are other datasets with similar sample sizes (e.g. ImageNet), the datasets we use have the advantage that they are low-resolution and an easier task— thus, scaling-law experiments are far more computationally feasible. We also do some experiments on a synthetic dataset in Appendix E.

**Architectures.** We use the following base architectures: Myrtle CNN (Page, 2018; Shankar et al., 2020) for the CIFAR-5m-bin task and a 5 layer CNN with 64 channels for the SVHN-parity task. We consider various width scaling for these networks: For the Myrtle CNN we vary the width from 16 to 1024 and from 16 to 4096 for the 5 layer CNN. See Appendix B for more details.

**Experimental Details.** We describe some subtleties in the experimental setup. We use NTK parameterization (Jacot et al., 2018) for both the neural network and the kernels as this is the parameterization used in proving the equivalence of neural network and NTK at infinite width. We train with MSE loss and $\pm 1$ labels. We use test error as the metric for all the plots except in Appendix I where we recreate some of the most important plots for test loss. All of our networks are in the overparameterized regime i.e. are able to reach 0 train error. To preserve the correspondence between the neural networks, empirical NTKs and infinite NTKs we train all of them with SGD with momentum with the same hyperparameters (Appendix B.4). This also ensures that in all experiments neural networks will be trained below the *critical learning rate*, i.e. the learning rate at which training of the empirical and infinite NTK can converge (Appendix B.3). Training for the empirical NTK is done by linearizing the initialized neural neural network using Novak et al. (2020) library while for infinite NTK we directly use SGD with momentum on the linear system given by the infinite NTK and the labels. We describe further experimental details for each individual experiment in Appendix B.

# 3 DATA SCALING LAWS OF NEURAL NETWORKS AND NTKS IN THE OVERPARAMETERIZED REGIME

In this section we compare the data-scaling laws of neural networks to their corresponding emperical NTKs and infinite NTKs. Our main claim is the following.

**Claim 3.1.** *There exists a natural setting of task and network architecture such that the neural network trained with SGD has a better scaling constant than its corresponding infinite and empirical NTK at initialization. Further, this gap in scaling continues to hold over a wide range of widths and learning rates used in practice.*

The above claim can be interpreted as stating that there exists natural settings where the regime in which real neural networks are trained is meaningfully separated from the NTK regime, and real neural networks have a better scaling law.

In Figure 1 (A), we train a Myrtle CNN (Page, 2018; Shankar et al., 2020), its empirical NTK at initialization, and its infinite NTK on the CIFAR-5m-bin task. In each case, we train to fit the train set with SGD and optimal early stopping. We then numerically fit scaling laws, and find, scaling-exponents $\beta$ of: .185 (empirical NTK), .213 (infinite NTK), .291 (neural network). Thus, in this image-classification setting, the real neural network significantly outperforms its corresponding NTKs with respect to data-scaling. We show the statistical significance of this result in Appendix A. See the Appendix B for full experimental details.

We now investigate how robust this result is to changes in the width of the architecture and optimizer, within realistic bounds.

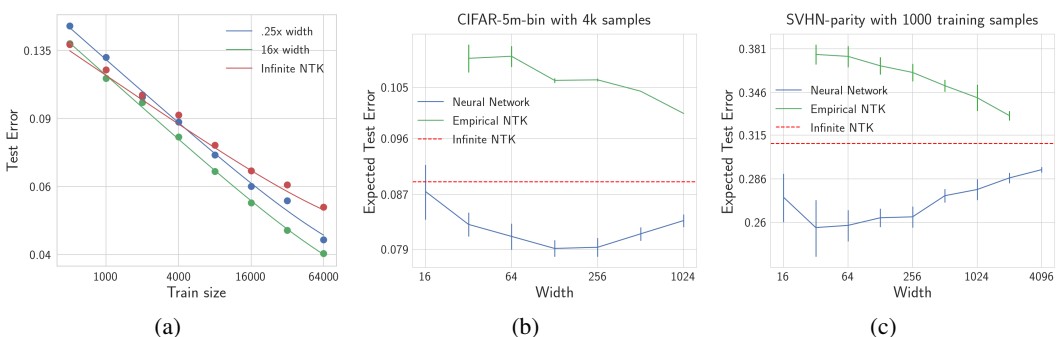

(a)             (b)             (c)

Figure 2: On the Effect of width. In Figure 2(a) we plot data scaling laws of the Myrtle-CNN at small (16) and large (1024) widths and its the infinite NTK. We observe that both finite widths have similar scaling constant which is better than that of the infinite NTK. In Figure 2(b) we plot the performance of Myrtle-CNN and its empirical NTK for a fixed training size while varying width. In Figure 2(c) we do the same for a 5-layer CNN and the SVHN-parity task. Both Figure 2(b) and 2(c) we observe that (a) the empirical NTK performance continues to improve with width, moving towards the infinite NTK performance while (b) neural network performance improves initially and then starts to deteriorate towards the infinite NTK performance. Error bars represent estimated standard deviation. See Appendix B for more details.

**Effect of Width.** We explore the effect of width. In Figure 2(a) we train neural networks with widths much smaller (16) and much larger (1024) than the width (64) used in Figure 1(A). We find that these networks behaved similarly with respect to their scaling constants (.276 and .279 respectively), and performed better than the infinite width NTK (scaling constant: .213), confirming that *real neural networks are far from the NTK regime*. However, we know that in the truly infinite width limit, all these methods will perform identically. Moreover, as mentioned in Section 2 we are careful to ensure this limit is preserved by our optimization and initialization setup. This implies that at some point, increasing width of the real network will start to *hurt* performance— although it may be computationally infeasible to observe such large widths. To explore the width-dependency, in Figure 2(b) we plot the expected performance of empirical NTK and Neural network as we *increase the width*, using a fixed training size of 4000. Here we see that (a) the empirical NTK at initialization

continues to improve with larger width and approaches the infinite NTK's error from above, while (b) the neural network improves initially and then starts to deteriorate and approach the infinite NTK's error from below. In Figure 2(c) we repeat the experiment for the SVHN-parity task. In this setup it was computationally feasible to try out much larger width (upto 4096) with a smaller training size of 1000. Hence in this experiment as the width increases, we can observe stronger deterioration of the performance of neural network, towards the infinite NTK performance.

Together these results suggest that "intermediate" widths (not too large, not too small) are important for the performance of overparameterized neural networks, and any explanatory theory must be consistent with this.

**Effect of Learning Rate.** We now study how robust our results are to changes in the learning rate, within practically used bounds. Note that changing the learning rate only affects the neural network training, and does not affect any of their corresponding NTKs. In Figure 3(a) we train networks in the same setting as Figure 2, but with varying learning rates. We find that after moderate modifications of the the learning rate the neural network still has a better scaling law than infinite and empirical NTK at initialization, suggesting that practically used learning rates (for practically used widths) are far from the NTK regime. The scaling constants are .333, .262, .213 for the 3x higher learning rate, 10x lower learning rate and the infinite neural network. We discuss the effects of more drastic changes (1000x) in the learning rate in Appendix C.

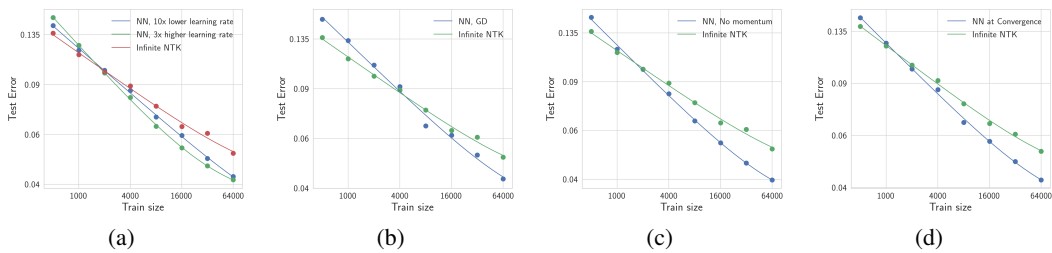

(a)                (b)                (c)                (d)

Figure 3: **Neural networks continue to have a better scaling constant under various hyperparameter choices.** We compare the data-scaling for a Myrtle-CNN, its empirical NTK and its infinite NTK on CIFAR-5m-bin task under various hyperparameter changes: (a) Higher and lower learning rate compared to Figure 1 (b) GD instead of SGD (c) SGD without momentum (d) Training until convergence (no early stopping)

**Other Changes in Optimization.** We now study whether our results hold under other changes to optimization parameters. In Figure 3(b), 3(c), 3(d), we see the effect of doing GD instead of SGD, effect of training without momentum and of using the final test error instead of doing optimal early stopping respectively. We see that in all of these cases, while there is some change in the scaling laws, the neural network scaling constant is still always better than the one for infinite NTK. The scaling constants for neural networks in Figure 3(b), 3(c), 3(d) are .294, .310 and .292 respectively. The scaling constant for the infinite NTK is .213 in Figure 3(b), 3(c) and .219 for Figure 3(d). This suggests that these optimization factors (within commonly used values) are not the fundamental reason behind the improved scaling laws of neural networks.

Various extensions to the NTK regime have been proposed (Roberts et al., 2021; Andreassen & Dyer, 2020) in the literature which allow for the change in empirical NTK but posit that higher order analogues of the NTK remain constant. This would predict that higher order analogues of the empirical NTK at initialization would be sufficient to match the performance of neural networks. In Appendix D we show that this is not the case suggesting that these theories may also not be sufficient to explain the performance of practical neural networks.

**Discussion and Future Questions.** The equivalence between neural networks and corresponding NTKs applies when width $\gg$ train-size. On the other hand nearly all overparameterized networks and natural tasks fall in the regime of width $\ll$ train-size (though width is still large enough to fit the dataset). The results of this section — showing separations between neural networks in the latter regime and NTKs lead to following concrete question on the gap between theory and practice which could guide future work.

**Question 3.1.** *How can we understand the the behavior of overparameterized networks in the width $\ll$ train-size regime?*

## 4 EXPLORATION OF AFTER-KERNEL WRT DATASET SIZE

In the previous section, we studied the empirical NTK when linearized around weights at *initialization*. In this section we will study the behaviour of empirical NTK when linearized around the weights obtained at the *end of training*. This is known as the *after-kernel* for empirical NTK, in the terminology of Long (2021). We will show, in the more precise sense defined below, that (1) the after-kernel continues to improves with dataset size, and thus (2) no fixed-time after-kernel is sufficient to capture the data scaling law of its corresponding neural network.

Formally, denote the after-kernel from the neural network trained on $m$ samples as $K_m$. We will denote the accuracy of $K_m$ when fit on $n$ samples as $K_m(n)$. Here, the $n$ samples are a subset of the original $m$ samples. When we use fresh $n$ samples to fit we use the notation $K_m^F(n)$. We study the after-kernel as improved performance of neural networks over NTKs has been attributed (Ortiz-Jiménez et al., 2021; Atanasov et al., 2021) to the adaptation of the empirical NTK of the neural network to the task. Concretely, prior works (Long, 2021; Paccolat et al., 2021) have shown that this explanation is complete in the following sense: The behaviour of $K_n(n)$ is similar to that of the neural network fit on $n$ samples. In other words, when we fit an after-kernel obtained from training on $n$ samples to the same $n$ samples we get an accuracy very close[1] to that of the neural network fit on the same $n$ samples. We verify this for our setup in Figure 4(a). This tells us that the following two factors are sufficient to explain the behaviour of neural networks fit on $n$ samples: (1) Change in empirical NTK from empirical NTK around initial weights to the after-kernel due to training on $n$ samples. (2) Fitting the after-kernel on $n$ samples.

What this does not tell us is how these two improvements scale with training size $n$. In particular, we know that $K_0(n)$ i.e. the empirical NTK at initialization fit on $n$ samples does not match the neural network trained on $n$ samples on the other hand $K_n(n)$ does. This raises the following natural question: *How data dependent does the kernel need to be to recover the performance of the neural network?* For example, it is possible that for some sample size $m_0$ and all $m \geq m_0$, the after-kernel $K_m$ is roughly constant, and has same scaling law as the neural network itself. We find that this is not the case– the after kenel continuously improves with dataset size $m$.

### 4.1 EXPERIMENTAL RESULTS

**After-Kernel continues to improve with dataset size.** In Figure 4(b) we plot $K_m(500)$ versus $m$ for our base Myrtle CNN from Figure 1. We observe that $K_m(500)$ improves as $m$ goes from 500 to $1024k$ showing that the after-kernel keeps improving with larger dataset sizes. Corresponding SVHN-parity experiments can be found in Appendix F.

**Fixed after-kernel is not sufficient to capture neural network data scaling.** In Figure 4(b) we plot the data scaling curves for the base Myrtle-CNN, its empirical NTK at initialization, $K_{16k}$, $K_{64k}$ with scaling constants $.291, .185, .103, .097$ respectively. We find that the neural network has the best scaling constant. This shows that the scaling of the after-kernel with training size is an important component of neural network scaling laws as even the after-kernel learnt with 64k samples (on the simple CIFAR-5m-bin task) is not sufficient to explain the data scaling of neural networks. We also see that $K_{64k}$ has better performance than $K_{16k}$, another evidence towards the fact that after-kernel improves with dataset size.

## 5 TIME DYNAMICS

In the previous section, we saw that the change in the empirical NTK from initialization to the end of training (the *after-kernel*) is sufficient to explain the improved performance of neural networks. Thus the empirical NTK must have evolved throughout training, and in this section we take a closer

---

[1]We again note that empirical NTK *does not* refer to the linearization of the network (See Section 2 for an exact definition). If we had linearized the network this statement would be trivially true as the linearized network around final weights would start out with an accuracy matching that of the trained neural network.

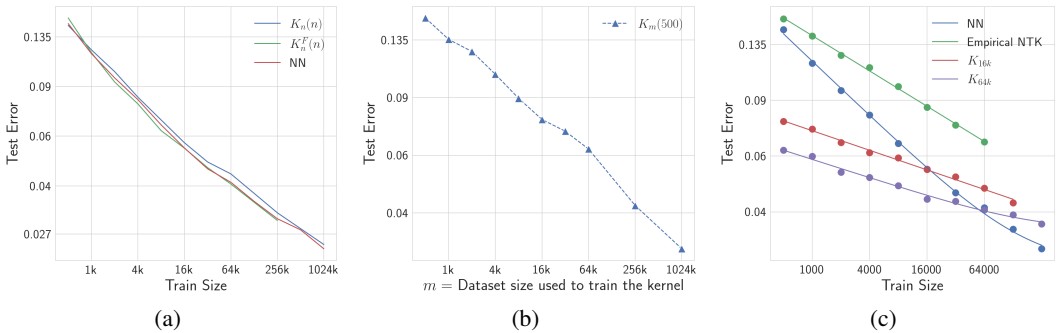

(a)  (b)  (c)

Figure 4: **After-Kernel continues to improve with dataset size.** In Figure 4(a) we plot data scaling curves of $K_n(n), K_n^F(n)$ and the neural network and observe that they behave very similarly. In Figure 4(b) we plot $K_m^F(500)$ versus $m$ and observe that the performance improved with increasing $m$. In Figure 4(c) we plot data scaling curves of empirical NTK at initialization, $K_{16k}$, $K_{64k}$ and the neural network. We observe that the neural network has the best scaling law amongst these.

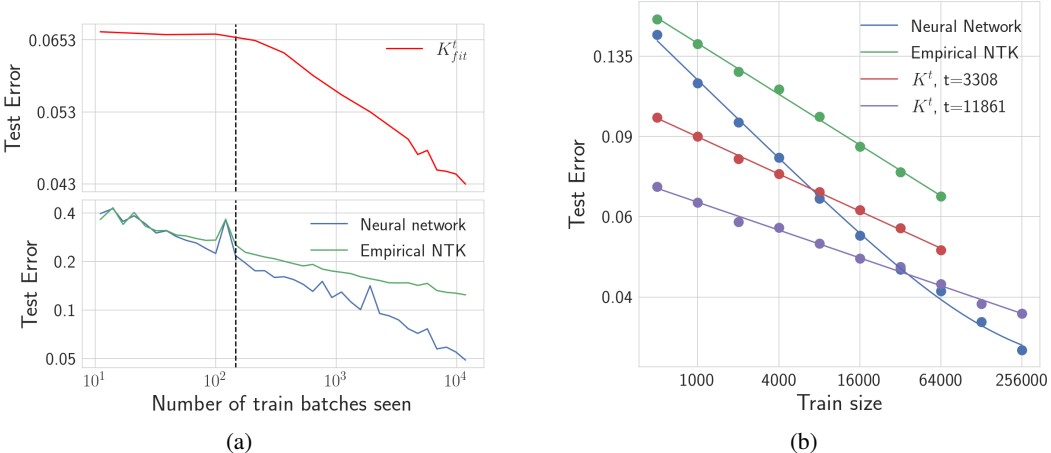

(a)  (b)

Figure 5: **Empirical NTK keeps improving uniformly throughout most of the training.** In Figure 5(a) we plot the test error of Myrtle-CNN, its empirical NTK at initialization and $K_{fit}^t$ at time $t$. We observe that the slope $K_{fit}^t$ does not decrease with time suggesting that the change in kernel does not slow down after an initial part of training. Using this same setup, we plot the data scaling curves of $K^t$ for various $t$ and the data scaling of Myrtle-CNN in Figure 5(b). We observe that the Myrtle-CNN has the best scaling law.

look at this evolution. Our main focus in this section is to investigate the following informal proposal in the literature (Fort et al., 2020; Long, 2021) about how the empirical NTK evolves:

**Hypothesis 5.1** ((Informal, from Fort et al. (2020); Long (2021)). *The empirical NTK evolves rapidly in the beginning of training ($< 5$ epochs), but then undergoes a "phase transition" into a slower regime.*

One way to interpret the above hypothesis is that there is both a qualitative and quantitative difference in the empirical NTK between the "early phase" of training (the first few epochs) and the later stage of training. This is called a "phase transition" in the literature, in analogy to physics, where systems undergo discontinuities between two regimes with quantitatively different dynamics.

In this section we will give evidence that suggests, contrary to prior work, that there is no such "phase transition". We show that if empirical NTK performance is measured at the appropriate scale, performance appears to continuously improve throughout training (from early to late stages), at approximately the same "rate." Our experiments are in fact compatible with the experiments in

prior work (e.g. Fort et al. (2020)): we simply observe that if performance and time are measured on a log-log scale (as is appropriate for measuring multi-scale dynamics), then the NTK is seen to improve continuously throughout most of the training.

## 5.1 EXPERIMENTS

**Setup.** We now describe the setup more formally. Let $K^t$ refer to the empirical NTK (as described in Section 2) extracted at time $t$ in the training, where we measure time in terms of number of SGD batches seen in optimization. $K^t_{fit}$ denotes the model $K^t$ fit to the whole training data. Prior works (Fort et al., 2020; Long, 2021) have used the slope of the curve of test error of $K^t_{fit}$ versus $t$ to decide if the kernel is changing rapidly or not. We will do the same with one crucial difference: We will measure this slope on a log-log plot instead of directly plotting test error and time. We do this as empirically scaling laws with respect to time (or tokens processed) have been observed (Kaplan et al., 2020) for natural language tasks in neural networks and formally proved for kernels (Velikanov & Yarotsky, 2021) for natural tasks. These results suggest the need for log-log plots to observe qualitative phase transitions in training dynamics.

**Results.** Our main claim is that the test error of $K^t_{fit}$ as a function of time $t$ is approximately linear on a log-log scale, throughout the course of training. Recall that $K^t_{fit}$ is the model obtained by extracting the empirical NTK after $t$ batches of training the real neural network.

In Figure 5(a) we compare the test error of the base Myrtle-CNN at time $t$, test error of empirical NTK at initialization at time $t$ and $K^t_{fit}$ when trained on $64k$ samples with the same hyperparameters as Figure 1. Since we want to probe Hypothesis 5.1, which is about the beginning of training, we plot these quantities until train error reaches $< 5\%$ (which requires 32 epochs in our experiments). This should be sufficient to cover any reasonable definition of "beginning of training".

Observe that in Figure 5(a) we do not observe a "phase transition" after which the improvement in kernel test error (in red) slows down. In fact, we observe that the kernel starts out being essentially constant and then starts and continues to improve uniformly.

We instead observe the following two regimes: **(1.)** In the first regime (before the dashed vertical line) the empirical NTK at initialization and the neural network have very similar behaviour, and $K^t_{fit}$ is nearly constant. This only lasts for around 140 batches $\approx 0.5$ epochs[2]. **(2.)** In the next regime (after the dashed line) the empirical NTK at initialization and the neural network diverge. As they diverge, the extracted kernel $K^t_{fit}$ also starts to improve with a constant slope, and this improvement continues uniformly until the terminal stage of training.

Importantly, the kernel $K^t_{fit}$ does not transition into a "slower phase" of learning at any point[3] in our experiments. Corresponding SVHN experiments can be found in Appendix F.

Next, we measure the performance of $K^t$ in terms of its *data scaling law*. Due to computational limitations (since measuring data-scaling is expensive), we can only measure the scaling law for several selected values of time $t$, instead of every batch (as in Figure 5(a)). In Figure 5(b) we plot data-scaling of $K^t$ for $t = 0$ (empirical NTK at initialization), 3308 (10 epochs), 11861 (32 epochs), in the same setup as Figure 5(a). We also plot the data scaling of the base Myrtle-CNN with the same hyperparameters. As in Figure 4(c) of Section 4 we again observe that the neural network has the best scaling law, outperforming any of the extracted kernels. This shows that representations learnt after any constant time $t$ of training are not sufficient to explain the data scaling of neural networks. Rather, these representations improve throughout training, and the entire course of training must be considered to recover the correct scaling law.

## REFERENCES

Ben Adlam and Jeffrey Pennington. The neural tangent kernel in high dimensions: Triple descent and a multi-scale theory of generalization. In *Proceedings of the 37th International Conference on Machine Learning, ICML 2020, 13-18 July 2020, Virtual Event*, volume 119 of *Proceedings of*

---

[2]Note that this means that if we plot per epoch this phase would not be visible at all.

[3]As we keep training at some point train loss will tend to 0 and $K^t_{fit}$ will converge to a fixed value. This does not affect our results as we are only interested in the initial part of training as described in Hypothesis 5.1

