# OpenReview forum: "Limitations of the NTK for Understanding Generalization in Deep Learning"
_ICLR.cc/2023/Conference — Submitted to ICLR 2023_

### Official Review · Reviewer_8CKY · 2022-10-23

**Confidence:** 4
**Clarity, Quality, Novelty And Reproducibility:** 1. In general the paper is easy to fo…
**Correctness:** 4
**Technical Novelty And Significance:** 2
**Empirical Novelty And Significance:** 3
**Recommendation:** 6

**Strength And Weaknesses:**

**Strengths**

1. Studying the difference between finite and infinite width through scaling laws is a very interesting and timely contribution. While prior work has focused on pointwise comparisons, scaling laws indeed provide a better assessment, which is lacking in the literature to the best of my knowledge. The result that neural kernels exhibit weaker scalings is a stronger one than what was previously known and further highlights the limitations of these approaches.

2. The authors have performed a very rigorous analysis by going beyond infinite-width and Empirical NTK at initialization. The results on the after-kernel and the fact that neither “early-stopped” nor weaker pre-trained ENTKs can catch up in terms of scaling is somewhat surprising and differs from earlier works.

3. The results in Figure 2b), 2c) are very interesting as well as it contradicts the wisdom that overparametrization always helps. There are some prior results on this phenomenon, for instance [1], referred to as triple descent. Do you observe the same phenomenon if the usual NTK assumptions are broken (e.g. large learning rate, standard parametrization etc)?

**Weaknesses**

1. The usage of optimal stopping is a bit unfair here as the infinite width NTK and the empirical NTK cannot profit from it (I assume you always consider kernels evaluated at t=inf?). Figure 3d) addresses this concern to some degree but I still think the results would be more representative if a certain computational budget would have been fixed before hand.
2. The paper does have a few typos (see clarity section) which makes reading the work a bit less enjoyable. It is also entirely missing a conclusion and discussion section, which is a bit of a shame as a nice summary of the results obtained in this work would really drive the message home.

[1] The Neural Tangent Kernel in High Dimensions: Triple Descent and a Multi-Scale Theory of Generalization, Ben Adlam, Jeffrey Pennington

**Summary Of The Paper:**

The authors investigate the generalization performance of convolutional neural networks and their associated kernels, i.e. the infinite-width neural tangent kernel, the empirical neural tangent kernel at initialization, as well as the “after-kernel”, i.e. the empirical tangent kernel arising after training the finite-width neural network for a certain amount of epochs. The authors assess this performance through so-called scaling laws, a metric that has been popular recently, which measures how test accuracy evolves as sample size is increased. The resulting curves traditionally follow a power law with the exponent indicating how well a model class generalizes. The authors identify that finite-width networks exhibit a better scaling compared to infinite-width kernels as well as empirical kernels both at initialization and after training. Contrary to other works, the after-kernel is found to improve even in later stages of training, stopping the empirical kernel from catching up to the finite-width network.

**Summary Of The Review:**

In summary, I’m leaning towards accepting this paper because studying neural kernels through scaling laws is an important contribution that has been absent in the literature. The results are very interesting and further highlight the limitations of neural kernels on a stronger level. Especially the findings on the after-kernel and its limitations are very novel and unexpected and should be of interest to the ICLR community. The paper should be improved however, the results should be properly discussed in the end and some more background on scaling laws and their role in explaining generalization of a model class should be provided.

---

### Official Review · Reviewer_mmPh · 2022-10-25

**Confidence:** 4
**Correctness:** 4
**Technical Novelty And Significance:** 2
**Empirical Novelty And Significance:** 4
**Recommendation:** 8

**Clarity, Quality, Novelty And Reproducibility:**

The paper is clearly written, results are novel, and the methodology and analysis is of high quality. The paper does provide a lot of details on the experimental setup. For reproducibility, however, it would have been great if the authors published their code as well.

**Strength And Weaknesses:**

Strength:
- interesting and relevant analysis
- solid, sound, and quite comprehensive analysis
- insightful results

Weakness:
- only evaluated for binary classification
- source code of experiments not published

**Summary Of The Paper:**

The paper empirically evaluates how well the NTK and empirical NTK approach captures the performance of neural networks. For that, it analyses the scaling behavior of the kernels in comparison to the original network wrt. dataset size and network width. The paper finds that neither NTK nor empirical NTK can capture the scaling behavior of neural networks and shows that this result is robust to changes in learning parameters. Moreover, the paper shows that the after-kernel also does not capture the scaling behavior of neural networks. This indicates that the NTK framework has significant limitations in capturing the performance and learning dynamics of neural networks in practice.

**Summary Of The Review:**

The paper presents an interesting empirical study on the abilities of the NTK framework to capture the performance and training dynamics of realistic neural networks. The experiments cover a range of approaches (NTK, empirical NTK, after-kernel) and analyze them from a wide range of perspectives. The results are insightful and relevant to the community. For future work it would be interesting to see whether these results hold for other learning tasks as well.

Since for such an empirical study reproducibility is particularly important, I would ask the authors to publish their code as open source (for the reviewing process this can be done, e.g., via an anonymized github https://anonymous.4open.science/).

Detailed comments:
- the paper appears to use proper parenthetical citation in the intro, but then only use textual citation afterwards. I suggest using textual citations only when the citation is part of the sentence, as recommended e.g., by the APA style (cf. https://apastyle.apa.org/style-grammar-guidelines/citations/basic-principles/parenthetical-versus-narrative)

---

### Official Review · Reviewer_fuCi · 2022-10-26

**Confidence:** 4
**Correctness:** 3
**Technical Novelty And Significance:** 1
**Empirical Novelty And Significance:** 2
**Recommendation:** 3

**Clarity, Quality, Novelty And Reproducibility:**

The main contribution of this paper is to demonstrate the empirical performance gap between NTK method and the neural network. But there have been many papers[1,2,3] that state the empirical and limiting NTK methods performs worse on empirical datasets than neural networks and have many limitations like poor sample complexity.

Besides, the tool used by the this work is really simple, and there are few comparisons and analysis for the results except listing the experiment results that was somewhat expectable.

Hence I think the novelty and originality of this work are not good enough.

The clarity of the article is clear in general.

[1] On Exact Computation with an Infinitely Wide Neural Net
[2] On the Power and Limitations of Random Features for Understanding Neural Networks
[3] What can linearized neural networks actually say about generalization?


**Strength And Weaknesses:**

Strength:
- Compared the NTK method with the neural network in several different ways.
- Comprehensive experiments to demonstrate the gap between NTK method and NN.

Weaknesses:
- The methodology is lack of innovation.
- The author just piled up many experiment results, but made no effort to explain the reason either in intuitive or mathematical way.
- The only appearance of NTK related formula are its definitions. I think there should be more analysis or discussion about the NTK itself.
- This paper only provided one metrics, the test error, among different methods with different parameters. Some statistics such as the variance of multiple experiments may be helpful.
- Only one network architecture was involved in this work.

**Summary Of The Paper:**

This paper utilizes the scaling laws as the tool to try to show the gap between the infinite and empirical NTKs, and the neural networks.

The author compares the scaling of real networks to the scaling of NTKs in 4 ways using some experiments.
- Data scaling of initial kernel
- Width scaling of initial kernel
- Data scaling of  after-kernel
- Time scaling.



**Summary Of The Review:**

The novelty and originality of this work are not good enough both in the sense of methodology and conclusion. I think what we're most interested in is why neural networks perform better than NTK in many ways, instead of investigating and comparing the performance of different methods for some specific network architecture on a specific dataset, which is not that striking.

---

### Official Review · Reviewer_PzLb · 2022-10-28

**Confidence:** 3
**Correctness:** 2
**Technical Novelty And Significance:** 2
**Empirical Novelty And Significance:** 2
**Recommendation:** 5

**Clarity, Quality, Novelty And Reproducibility:**

Some questions:

1. I don’t think this statement “realistic settings where as the width of the neural network increases to very large values, the test performance of the network gets worse and approaches the performance of the infinite NTK, unlike existing results in literature which suggest that over-parameterization is always good” is correct. Please cite these works.

2. Can the authors elaborate on how this work helps understand “the extent to which understanding NTKs (empirical and infinite) can teach us about the success of neural networks”?

3. I find the introduction quite difficult to parse. It jumps right into discussing results without many important details about concepts such as “scaling behavior” and “data scaling component” as well as the experimental setup.

4. In both experiments, the tasks are binary classification. Why is MSE loss used instead of cross-entropy?


**Strength And Weaknesses:**

Strengths:
- This work provides another empirical confirmation that finite neural networks are superior to their corresponding NTK models, from the viewpoint of scaling laws.
- It demonstrates that even when trained with more samples, the empirical NTK obtained from a trained network still scales worse than the neural network.


Weakness:
- Scaling laws for neural networks and empirical NTKs were studied in the previous work, as cited in this paper. I also believe that infinite NTK models have been widely known to be inferior to finite neural networks.
- I don’t see much more values and insights in comparing data scaling exponent versus directly comparing test accuracy.
- Since NTKs are not realistic neural networks, obviously NTK approaches would not help in understanding the neural network generalization.

**Summary Of The Paper:**

This work studies the limitations of NTKs compared to finite neural networks properly trained with SGD. Through the lens of scaling laws, this paper shows that neural networks scale better (in terms of data scaling exponents) than both infinite and empirical NTK models. Additionally, it demonstrates some other properties, namely data scaling of after-kernel and time scaling for empirical NTKs.


**Summary Of The Review:**

See above.

---

### Decision · Program_Chairs · 2023-01-20

**Decision:**

Reject

**Justification For Why Not Higher Score:**

As I have written in the summary section, this paper's novelty is rather limited. If there were more novel insights or theories, this paper would have more value. The numerical experiments mainly focus on the test error. It is expected that there were more substantial (and interpretable) findings about the crucial factor that makes separation between NTKs and neural networks.

**Justification For Why Not Lower Score:**

N/A

**Metareview: Summary, Strengths And Weaknesses:**

This paper investigate limitation of Neural Tangent Kernel (NTK) by focusing on the *scaling law* of the infinite width NTK and empirical NTK against neural networks instead of point-wise comparison. It is shown experimentally that NTK has worse scaling law against neural networks, which indicates the limitation of NTK framework to explain the performance of neural networks.

Investigating the plausibility of NTK theory for justifying effectiveness of deep learning is indeed an important research topic. The authors conducted extensive experiments in which the scaling laws of several settings are investigated. These experimental results are valuable to the community.
However, there are also some weakness in this paper as follows.
- The main criticism raised by the reviewers is its novelty. Indeed, several previous work has pointed out superiority of neural network from a fixed kernel methods such as NTKs. Although the scaling law is the main focus of this paper, some numerical and experimental studies have been conducted to investigate the scaling laws as pointed out by reviewers PzLb and fuCi (in particular, several theoretical work claim that feature learning can make separation between kernel methods and neural networks in terms of its scaling property).
- Although several reviewers acknowledged the extensive experiments, they (especially, fuCi and 8CKY) also raised a concern that some more background and mechanism behind the observation in the numerical experiments could be discussed in more details.

Although the authors' efforts were appreciated by the reviewers, this paper provides rather limited novel insight as described above.
A reviewer initiated a discussion championing this paper, but the other reviewers did not support their suggestion. On the other hand, even reviewers who are positive on this paper consider that there is room for improve in the discussion deduced from the experiments.

For these reasons, the AC unfortunately cannot recommend acceptance of this paper to ICLR.